# Valorization of the Red Algae *Gelidium sesquipedale* by Extracting a Broad Spectrum of Minor Compounds Using Green Approaches

**DOI:** 10.3390/md19100574

**Published:** 2021-10-14

**Authors:** Natalia Castejón, Maroussia Parailloux, Aleksandra Izdebska, Ryszard Lobinski, Susana C. M. Fernandes

**Affiliations:** 1Universite de Pau et des Pays de l’Adour, IPREM, E2S UPPA, CNRS, 64600 Anglet, France; m.parailloux@univ-pau.fr (M.P.); aizdebska@univ-pau.fr (A.I.); ryszard.lobinski@univ-pau.fr (R.L.); 2MANTA—Marine Materials Research Group, Universite de Pau et des Pays de l’Adour, E2S UPPA, 64600 Anglet, France; 3Polymer Chemistry, Department of Chemistry—Angstrom Laboratory, Uppsala University, Lagerhyddsvagen 1, 75120 Uppsala, Sweden

**Keywords:** ultrasound-assisted extraction, eco-friendly methods, green extraction, macroalgae, bioactive compounds, mycosporine-like amino acids, phycobiliproteins

## Abstract

Until now, the red algae *Gelidium sesquipedale* has been primarily exploited for agar production, leaving an undervalued biomass. In this work, the use of eco-friendly approaches employing ultrasound-assisted extraction (UAE) and green solvents was investigated to valorize the algal minor compounds. The green methods used herein showed an attractive alternative to efficiently extract a broad spectrum of bioactive compounds in short extraction times (15 to 30 min vs. 8 h of the conventional method). Using the best UAE conditions, red seaweed extracts were characterized in terms of total phenolics (189.3 ± 11.7 mg GAE/100 g dw), flavonoids (310.7 ± 9.7 mg QE/100 g dw), mycosporine-like amino acids (MAAs) (Σ MAAs = 1271 mg/100 g dw), and phycobiliproteins (72.4 ± 0.5 mg/100 g dw). Additionally, produced algal extracts exhibited interesting antioxidant and anti-enzymatic activities for potential applications in medical and/or cosmetic products. Thus, this study provides the basis to reach a superior valorization of algal biomass by using alternative methods to extract biologically active compounds following eco-friendly approaches. Moreover, the strategies developed not only open new possibilities for the commercial use of *Gelidium sesquipedale,* but also for the valorization of different algae species since the techniques established can be easily adapted.

## 1. Introduction

Marine sources, especially seaweeds and microalgae are still an unexploited reservoir of bioactive compounds, which have significant potential to provide novel and natural ingredients for food and pharmaceutical industries [1,2].

Up to now, the red algae *Gelidium sesquipedale* has been mainly commercially exploited for agar production [3], leaving a large undervalued algal biomass. The industrial extraction process for agar generates extreme amounts of by-products that are basically used as fertilizer or frequently discarded since they are considered waste [4]. However, food processing by-products obtained from plants or algae are known as important sources of functional bioactive compounds [5,6]. Moreover, in the current context of global warming, the European Union (EU) is aiming to ensure the sustainability of the marine environments through its environmental and fisheries policies. The EU Blue Growth initiative represents a long-term strategy to support growth in the maritime sector as a whole by using the unexploited potential of oceans, seas, and coasts for economic growth [7]. A critical concern under this initiative is the valorization of algal biomass. EU objectives include not only the optimization of existing bioprocesses at the industry level, but also the quest for new products and/or environmental processes that improve the overall economic feasibility of algal biomass [8]. Thus, the search for new approaches that will successfully increase the value of algal biomass using minimum energy is nowadays a primary goal.

There are signs in the literature of the effective nutritional value and biological activities from *Gelidium sesquipedale* extracts, such as anti-enzymatic, antimicrobial, antioxidant, anti-inflammatory, or cytotoxic activities [9,10,11]. Moreover, the fatty acid composition, the total phenolic (TPC) and total flavonoid (TFC) contents, the identification of functional low-molecular-weight carbohydrates, the recovery of proteins and the detection of mycosporine-like amino acids (MAAs) have been reported in the literature [12,13,14,15,16], showing the potential of *Gelidium sesquipedale* biomass. Nevertheless, extraction methods commonly used for the isolation of these bioactive compounds are based on conventional techniques, which imply long extraction times, the use of high volumes of organic solvents, and high energy requirements, producing environmental and health problems [17].

Commercial interest in more sustainable and greener extraction approaches has increased during the past years, driven by growing consumer demands for more eco-friendly alternatives and natural ingredients that do not involve toxic chemicals. In this sense, ultrasound-assisted extraction (UAE) is a key technology in achieving the aims of sustainable chemistry and green extraction. Using UAE, selective extractions can be done in minutes with high reproducibility, reducing the consumption of toxic solvents, simplifying manipulation, and consuming only a fraction of the energy normally needed for the conventional solid–liquid extraction methods, such as Soxhlet extraction, maceration or distillation [18].

On the other hand, one of the most critical points in the extraction of bioactives from seaweeds is the selection of the extraction technique, due to the presence of a dense and firm cell wall [19]. For that reason, the seaweed cell wall must be properly disrupted to efficiently recover intracellular bioactive compounds [20]. In this regard, UAE could be the key to develop new and environmentally friendly extraction methods for seaweed biomass, due to its proven effective action in cell membranes and cost-competitive results [21]. UAE implies the use of ultrasound waves generated by a water bath or an ultrasonic probe which produce cell disruption and facilitates the extraction by the cavitation phenomenon. In recent years, ultrasound technology has been widely applied for green and economically viable extractions of valuable compounds from marine sources [22].

Therefore, the main objective of this study was to valorize the minor compounds of the red algae *Gelidium sesquipedale* by using eco-friendly extraction approaches in combination with green solvents. To extract the broad spectrum of bioactives, extraction conditions using ultrasound treatment with ethanol, water, and their mixtures as solvents were optimized to achieve green alternatives. Red seaweed extracts were characterized in terms of polyphenol, flavonoid, MAAs, and phycobiliprotein contents. Moreover, antioxidant properties and enzymatic inhibitory activities were evaluated by using in vitro activity assays. The results presented in this work will provide the basis for the development of alternative strategies to extract biologically active compounds following the principles of Green Chemistry to reach a superior valorization of algal biomass.

## 2. Results and Discussion

### 2.1. Total Phenolic and Flavonoid Contents of Red Seaweed Extracts

Polyphenols are one of the largest and most widely distributed groups of seaweed phytochemicals, which have gained special attention due to their pharmacological activity and health-promoting benefits [23]. Even though it is known that red seaweeds are not the main source of phenolic and flavonoids compounds, these metabolites are targeted compounds for the valorization of the minor compounds from *Gelidium sesquipedale*.

Figure 1 shows the effects of UAE treatment time (15 and 30 min), temperature (RT and 40 °C), and solvent used (ethanol, water, and aqueous ethanolic solutions (50% and 70% *v*/*v*)) on total phenolic (Figure 1a) and total flavonoid (Figure 1b) contents in comparison with the traditional extraction method.

Significant differences were found in the TPC between the different extraction conditions and solvents used, ranging from 41.5 to 252.2 mg GAE/100 g dw. Among the solvents investigated, ethanol:water (50:50 *v*/*v*) was the most effective solvent for the extraction of TPC except for the extracts produced with ultrasounds at room temperature, where water was also shown to be a good solvent to extract polyphenols (Figure 1a). The highest phenolic content was achieved with the conventional method using ethanol:water (50:50 *v*/*v*) (252.2 ± 7.3 mg GAE/100 g dw) (*p* < 0.05 in comparison with UAE treatments). Regarding the studied UAE conditions, no significant differences (*p* > 0.05) were found for the extracts obtained using ethanol:water (50:50 *v*/*v*) at 40 °C during 15 and 30 min (189.3 ± 11.7 mg GAE/100 g dw and 205.6 ± 7.7 mg GAE/100 g dw, respectively). Therefore, UAE at 40 °C for 15 min is a less time-consuming alternative to extract TPC. Even though the UAE approaches proposed in this study did not achieve the maximum recovery of TPC obtained by the traditional method, the application of ultrasounds could be an attractive alternative due to the shorter extraction time (15 min vs. 8 h), obtaining an extraction efficiency of 81% (extraction efficiency calculated considering the maximum TPC extracted with the traditional method).

A more interesting finding was observed during the extraction of flavonoids. UAE was found to be more effective in the extraction of flavonoid compounds (Figure 1b). Compared to the traditional method, ultrasound treatment significantly improved the extraction efficiency of flavonoids (*p* < 0.05). Among the solvents investigated, ethanol:water (70:30 *v*/*v*) showed the highest recovery of flavonoid compounds, with 40 °C and 30 min being the best conditions to extract these bioactive compounds from the red algae (310.7 ± 9.7 mg QE/100 g dw) (*p* < 0.05 for all the conditions tested in this study). Using the optimized UAE conditions, the content of flavonoids increase 1.3 times in comparison with the traditional method, showing the potential of ultrasounds to extract flavonoid compounds from algal biomass in a short time. These results were in agreement with other authors that also reported the use of ethanol:water (70:30 *v*/*v*) as the best solvent mixture to extract flavonoids from different vegetal sources [24,25,26] and seaweeds [27]. Similar results were obtained by Ummat et al. [28], who studied the TPC and TFC of 11 seaweed extracts from different species using the conventional solvent extraction and UAE. The authors also concluded that UAE was more effective than the traditional extraction in terms of total phenolic and flavonoid contents.

Other authors have already investigated the extraction of phenolic and flavonoid compounds from *Gelidium sesquipedale*, however using organic and toxic solvents and/or traditional extraction methods. For instance, Metidji et al. [29] reported a phenolic content ranging from 3.49 to 101.05 mg GAE/g dried extract and a lower content of flavonoids (ranging from 0.85 to 5.63 mg QE/g dried extract) using different mixtures of organic solvents like chloroform, methanol, diethyl ether or n-butanol. In another study, Grina et al. [10] described the phenolic and flavonoid contents of *Gelidium sesquipedale* extracts, but using higher temperatures than in the present study. Using the classical solvent extraction method with 70% ethanol at 60 °C for 2 h, the authors described a phenolic content of 11.1 ± 0.03 µg pyrocatechol equivalents (PE)/mg extract and a flavonoid content of 5.84 ± 0.02 µg QE/mg extract. Matos et al. [13] also studied the phenolic content of *Gelidium sesquispedale* using ethanol (86 ± 6 mg GAE/100 g dw) and water (70 ± 6 mg GAE/100 g dw) as solvents; however, the authors did not detail the extraction conditions or method used. Moreover, Xu et al. [30] described a phenolic content of 16.2 ± 1.0 mg GAE/g dry extract using ethanol but applying longer extraction times (24 and 48 h) than in the present study. Therefore, the alternative strategies reported in this study showed clear advantages for the extraction of phenolic and flavonoid compounds from *Gelidium sesquipedale* using green approaches and eco-friendly solvents in shorter extraction times.

### 2.2. Profiling of Mycosporines and Mycosporine-like Amino-Acids (MAAs) in Red Seaweed Extracts

Mycosporines and MAAs are a large family of natural molecules, which have exceptional ultraviolet-absorbing capacities. They are perfect candidates to produce high-value products due to their high abundance in red algae and extraction facility, remaining easily transposable at an industrial scale [31,32,33]. Moreover, these metabolites have been the subject of numerous researches in cosmetic and biomedical fields because of their potent photoprotective and antioxidant bioactivities [34,35,36].

As a first attempt to evaluate the best strategy to extract mycosporines and MAAs from *Gelidium sesquipedale*, the UV-visible absorption spectra of red seaweed extracts were analyzed (Appendix A from the Appendix A). Results suggested that water was the most promising solvent to recover MAAs with high purity since the other solvents used also showed the presence of other minor compounds. Therefore, water extracts produced by UAE approaches and the traditional method were selected to analyze the MAAs profile and their quantification using advanced methods (Table 1).

The screening of MAAs was performed using an untargeted ddMS²/MS^3^ analysis allowing an annotation of candidate-compounds based on a set of fragment ions, and neutral and radical losses specific to their fragmentation pathways acquired in positive ionization mode [15]. Data-processing using an untargeted workflow on Compound Discoverer^TM^ software allowed the inventory of tens of putative-MAAs with a signal intensity greater than 1 × 10^5^ and more than four characteristic fragment ions in their HCD70 MS² spectra (Table 1, Appendix A from the Appendix A). The identified compounds belonged mainly to the palythine and glycine families, thus confirming their predominance in red macroalgae [37]. Interestingly, two *m*/*z* values (*m*/*z* 231.1340 and *m*/*z* 305.1342) were reported as unknown compounds (Appendix A from the Appendix A). The detection of seven characteristic fragment ions in their HCD70 MS² spectrum confirmed their affiliation to the MAAs compound class. Notably, there is no significant difference between the different extraction procedures in the MAAs distribution, except for the extracts produced by UAE at 40 °C for 30 min, for which only eight compounds showed a signal above the defined intensity threshold.

In parallel, the extraction efficiency of the different UAE procedures was determined by the quantification of the most predominant MAAs: porphyra-334, shinorine, palythine and asterina-330 (Table 2). The quantitative study was based on a standard addition method with purified forms of palythine, shinorine and porphyra-334. The concentration of asterina-330 was determined using a semi-quantitative strategy based on the electrospray MS response factor similar (within 10%) to the available standards for which it displays structural relationships [15].

The results showed the MAAs yield of the different UAE procedures varied as a function of the temperature and the duration of the extractions. The algal extracts produced by UAE for 15 min at room temperature yielded the greatest MAAs concentrations. The efficiency of this alternative method was comparable to those obtained with the traditional procedure (∑ MAAs 1271 ± 352.9 mg/100 g dw and 1372 ± 253.8 mg/100 g dw, respectively). Interestingly, extraction yields using UAE for 15 min was even higher (*p* < 0.05) than those achieved with UAE for 30 min (∑ MAAs 1372 ± 253.8 mg/100 g dw and 886 ± 74.2 mg/100 g dw, respectively). Optimization of the extraction yields by applying temperature has been also investigated. Results showed that an increase of temperature resulted in a significant decrease of MAAs content (*p* < 0.05). Moreover, extraction yields obtained by UAE for 15 min at 40 °C were ten to 100-fold higher than UAE for 30 min at 40 °C, assuming MAA instability in a long-term temperature application.

In literature, the quantification of MAAs in *Gelidium sesquipedale*-based eco-friendly extraction methods have not yet been reported. A prior study dealing with quantification of MAAs in *Ptercladiela capillacea* and *Gelidium amansii* employed classical extractions based on methanol 25% (*v*/*v*) for 2 h at 45 °C and determined shinorine as the most predominant in *Gelidium amansii* [38]. Therefore, the application of ultrasounds for 15 min can be considered as a fast and valuable alternative to extract MAAs in comparison to the conventional method, which usually involves multistep processes. Interestingly, asterina-330 showed the greatest amounts in all extraction conditions, which assumed its specificity to the algal species *Gelidium sesquipedale* [31].

### 2.3. Antioxidant Activity of Red Seaweed Extracts

The antioxidant capacity of red seaweed extracts produced by UAE and conventional extraction methods was measured using 1,1-diphenyl-2-picryl-hydrazyl (DPPH) and ferric reducing antioxidant power (FRAP) assays (Figure 2).

There was a statistically significant influence observed between extraction conditions and solvent type used (*p* < 0.05). The strongest DPPH free radical scavenging activity was shown by the extract produced by UAE at 40 °C during 30 min (87.7 ± 4.8 mg TE/100 g dw) and the traditional extraction method (86.2 ± 3.9 mg TE/100 g dw) using in both cases ethanol:water (70:30 *v*/*v*) as the solvent (*p* > 0.05), while the highest FRAP value was shown by the extract obtained using UAE (115.3 ± 5.9 mg TE/100 g dw) with the same extraction conditions previously described (*p* < 0.05 in comparison with the traditional method). In contrast with the results achieved by the traditional method, the extract produced using ethanol:water (50:50 *v*/*v*) by UAE at 40 °C during 30 min exhibited similar DPPH (79.8 ± 10.1 mg TE/100 g dw) and FRAP (112.0 ± 3.6 mg TE/100 g dw) values than ethanol:water (70:30 *v*/*v*) (*p* > 0.05); consequently, the best antioxidant activity could be achieved using UAE with both ethanolic mixtures. Conversely, the weakest DPPH free radical scavenging activity (28.3 ± 1.7 mg TE/100 g dw) and the lowest FRAP value (46.8 ± 3.0 mg TE/100 g dw) was shown by the extract produced using the traditional method with pure ethanol.

Overall, the extracts produced using UAE showed similar or even higher (using the FRAP assay) antioxidant activity compared to the extracts obtained by the conventional method. It is known that the application of ultrasounds likely facilitated the release of bioactive compounds from seaweeds, and in consequence, extracts could exhibit a strong antioxidant capacity. Among the solvents tested, the aqueous ethanolic solutions (50% and 70% *v*/*v*) showed the most promising results. This was in agreement with the highest content of phenolic and flavonoid compounds shown in the present study, suggesting that it may be a positive correlation between the antioxidant activity and the TPC and TFC. A similar finding was reported by Chan et al., who described a positive role of the algal polyphenols extracted from the red seaweed *Gracilaria changii* as free radical scavengers and ferric ion reducing agents [39]. In a similar study, Zakaria et al. found synergistic effects of the phenolic compounds and the antioxidant capacity of the crude extracts of the red seaweed *Acanthophora spicifera* [40]. Moreover, Farasat et al. reported that the phenolic compounds, including flavonoids, are the main contributors to the antioxidant activity in different seaweed species [41].

Using ethanol and conventional solvent extraction, Xu et al. [30] reported the antioxidant capacity of different seaweeds, including *Gelidium* sp. collected from the coastline of eastern Guangdong in China. The authors showed a FRAP value of 16.1 ± 1.1 mg of gallic acid equivalent/g dry extract. In a similar study, Grina et al. [10] described the antioxidant activity of five seaweed species collected from the Moroccan Atlantic Ocean. The aqueous ethanolic extract (70% ethanol) of *Gelidium sesquipedale* produced by the traditional solvent extraction at 60 °C for 2 h were used to evaluate the antioxidant capacity with different methods: DPPH assay (IC_50_ = 84.61 ± 3.9 mg/mL), β-carotene-linoleic acid assay (IC_50_ = 75.36 ± 3.6 mg/mL), ABTS assay (IC_50_ = 44.46 ± 2.4 mg/mL) and FRAP (IC_50_ = 83.73 ± 2.9 mg/mL). Conversely, Matos et al. [13] evaluated the antioxidant activity of water and ethanol extracts from *Gelidium sesquipedale* using the same antioxidant method proposed in the present study; however, the authors did not show any activity for the water extracts in terms of DPPH or FRAP and only a DPPH inhibition of 6.8% for the ethanol extract. The scarce studies about *Gelidium sesquipedale* in the literature, the utilization of several methods to evaluate the antioxidant activity, and the different units used to express the antioxidant capacity make the comparison of the results difficult.

### 2.4. Extraction of Phycobiliproteins and Evaluation of Their Antioxidant Capacity

Phycobiliproteins are the most important component of light-harvesting complexes in cyanobacteria and red algae. Phycobiliproteins from red algae, namely R-Phycoerythrin (R-PE) and R-Phycocyanin (R-PC), are water-soluble red and blue pigments, respectively. Moreover, phycobiliproteins have a great potential in food, cosmetics, and medical applications due to their antioxidant, radical-scavenging, and anti-inflammatories activities [42]. Extraction of phycobiliproteins from microalgae has been extensively reported in the literature, being the cyanobacteria *Arthrospira Platensis* the most popular commercial source [43]. Red algae are also a rich source of phycobiliproteins and could be targeted compounds for the valorization of algal biomass due to the high price of these extracts in the market [44]. For that reason, in this work, the extraction of phycobiliproteins using UAE and maceration to produce high added-value products was investigated (Table 3). To the extent of our knowledge, this is the first time that the extraction of phycobiliproteins using ultrasounds from the red algae *Gelidium sesquipedale* has been studied.

To estimate the maximum amount of phycobiliproteins that could be extracted from *Gelidium sesquipedale* biomass, a serial extraction using five consecutive cycles of 1 h (5 h of total extraction time) was carried out, achieving a total phycobiliprotein content of 147.3 ± 3.1 mg/100 g dw. To develop alternative approaches to extract these valuable compounds, the application of ultrasound-assisted extraction was studied. For instance, two different extraction times were evaluated (10 and 15 min). However, longer extraction times were not evaluated due to the thermo-sensitive nature of phycobiliproteins. The total phycobiliprotein content of ultrasound extracts was 54.7 ± 1.6 mg/100 g dw and 54.1 ± 2.1 mg/100 g dw for 10 and 15 min, respectively, with a extraction efficiency of 37%, showing that the application of ultrasound was not able to extract the total amount of phycobiliproteins present in the algal biomass. In consequence, the combination of ultrasound and maceration was also investigated. This idea was taken for a previous published study of Mittal et al. [45], in which the authors examined the extraction of phycobiliproteins from *Gelidium pusillum* using maceration followed by the application of ultrasound treatment. However, it is known that phycobiliproteins are intracellular compounds; thus, cell disruption is necessary to achieve maximal efficacy during the extraction process [46]. For that reason, in this work, the use of ultrasounds as a pre-treatment followed by a maceration step to promote the release of phycobiliproteins into the medium was investigated. Specifically, a pre-treatment step using 15 min of ultrasound and two different maceration times (45 min and 1 h) was investigated. As can be seen in Table 3, the combination of both methods significantly improves the extraction of phycobiliproteins in comparison with the use of ultrasound alone (*p* < 0.05). However, increasing the extraction time of the maceration step from 45 to 60 min did not produce the expected impact, and the phycobiliproteins content improved only slightly (*p* < 0.05). These results suggest that the application of different cycles is a key condition to extract the total amount of phycobiliproteins present in red algae, probably related with the saturation of the solvent. Nevertheless, compared with previous data published, the results shown in this study were in agreement with the content of phycobiliproteins reported by other authors using different *Gelidium* species. For instance, Mittal et al. [45] reported a phycobiliprotein content of 20 to 200 mg/100 g dry biomass for *Gelidium pusillum* depending on extraction conditions. In another study, Sukwong et al. [47] reported a R-PE and R-PC content of 10.9 mg/100 g for *Gelidium amansii*.

Interesting results were found during the evaluation of the antioxidant capacity of produced extracts; the highest DPPH and FRAP values were exhibited by the extracts produced with the combination of ultrasound and maceration for 45 min (US 15 min + Mac 45 min) (*p* < 0.05). These results suggest that even the traditional extraction (serial extraction) was able to extract a higher content of total phycobiliproteins; the conventional method could induce the degradation of these compounds and consequently decrease the antioxidant activity of produced extracts (*p* < 0.05 for all the extracts tested). Therefore, the combination of ultrasound and maceration for 45 min seems to be the most promising strategy to extract the phycobiliproteins from *Gelidium sesquipedale* without losing their antioxidant capacity. Additionally, to further investigate the biological activity of phycobiliproteins, the extracts produced by the best conditions and the serial extraction to compare were evaluated in the following section.

### 2.5. Anti-Enzymatic Activities of Red Seaweed and Phycobiliproteins Extracts

In an attempt to find novel biological activities of *Gelidium sesquipedale* extracts, the inhibitory effects of produced extracts on acetylcholinesterase (AChE), tyrosinase, and elastase were evaluated. To the best of our knowledge, this is the first time that the enzymatic inhibitory activity of *Gelidium sesquipedale* extracts produced by ultrasound have been investigated. Moreover, data on this topic is scarce, and only a few studies have investigated the anti-enzymatic activities of *Gelidium sesquipedale* extracts.

In a first step, all red seaweed extracts produced were evaluated for their ability to inhibit the activity of different enzymes using a first screening approach (data not shown). However, the use of ethanol or its mixtures produced a negative effect on the activity of the enzymes investigated. For that reason, only the extracts produced by aqueous extraction (red seaweed extracts using water and phycobiliproteins extracts) were selected to evaluate their efficacy by measuring the half-maximal inhibitory concentration (IC_50_) (Table 4).

Acetylcholinesterase (AChE) is an enzyme that catalyzes the breakdown of acetylcholine to acetate and choline. It is also the potential target of most of the drugs used for the treatment of Alzheimer’s disease [48]. In this study, the AChE inhibitory activity of aqueous red seaweed extracts and phycobiliproteins fractions was evaluated. Moreover, neostigmine bromide was used as a positive control as it is used to treat Alzheimer’s patients [49]. As can be seen in Table 4, all seaweed extracts showed AChE inhibitory activity. Surprisingly, the aqueous extract produced by the traditional method was the most active among the other extracts investigated (*p* < 0.05). The anti-enzymatic activity decreased as the extraction temperature increased, exhibiting a negative correlation between the temperature and the activity of the extract. These results were in agreement with a previous study in which Grina et al. [10] reported a high IC_50_ value for a *Gelidium sesquipedale* extract (> 200 mg/mL) produced using high temperature (60 °C for 2 h).

Additionally, the potential of produced extracts for possible cosmetics products was investigated. Specifically, the inhibitory activity of red algal extracts towards two enzymes was evaluated: tyrosinase and elastase. Tyrosinase plays an important role in the biosynthesis of melanin. For that reason, the downregulation of tyrosinase is one of the most prominent approaches for the development of whitening and lightening products with applications in the cosmetic industry [50]. Conversely, elastase is a protease that reduces elastin in the skin by dividing specific peptide bonds. Inhibitors of elastase can be used as cosmetic ingredients to prevent loss of skin elasticity and thus skin aging [51]. In this work, the phycobiliproteins fractions were the most active extracts towards tyrosinase and elastase, while the extracts produced using water did not show anti-enzymatic activities towards these enzymes. Furthermore, a positive effect of the phycobiliprotein extracts produced with UAE was demonstrated, suggesting that ultrasound treatment may extract a broader spectrum of bioactives that could contribute to these activities. Grina et al. [10] reported a IC_50_ value of tyrosinase for *Gelidium sesquipedale* extracts as higher than 200 mg/mL. In another study, Oumaskour et al. reported an elastase inhibition greater than 95% with a dichloromethane-methanol (50:50 *v*/*v*) extract from *Gelidium sesquipedale*; however, the authors did not provide information about the IC_50_ value [52].

From the results gathered, it is possible to conclude that the water and phycobiliprotein extracts produced by the conditions used in this work exhibited a wide range of inhibitory activities, being promising extracts with potential anti-enzymatic activities for medical and cosmetic applications. Although aqueous extracts from *Gelidium sesquipedale* showed promising results and may open new opportunities for the exploitation of natural enzymatic inhibitors from marine resources, further studies are needed to clarify the identity of the metabolites responsible for these biological effects.

## 3. Materials and Methods

### 3.1. Materials

#### 3.1.1. Algal Biomass

Biomass of the red macroalgae *Gelidium sesquipedale* was provided by the Comité Interdépartemental des Pêches Maritimes et des Élevages Marins des Pyrénées Atlantiques et des Landes (CIDPMEM 64–40, Ciboure, France), which directly collected the algal biomass from the French Basque Coast at the end of September 2020. After the reception of the biomass, seaweeds were rinsed with tap water (to eliminate impurities), dried at room temperature for 2 weeks in a drying room, and milled using a laboratory ball mill (Planetary ball mill PM 100, Retsch GmbH, Germany). Samples were stored at 4 °C until their use.

#### 3.1.2. Chemicals

Tyrosinase from mushroom, L-3,4-dihydroxyphenylalanine (L-DOPA), elastase from porcine pancreas, N Succinyl-Ala-Ala-Ala-p-nitroanilide (AAAPVN), Acetylcholinesterase (AChE) from *Electrophorus electricus*, acetylthiocholine iodide, DTNB (5,5′-dithiobis (2-nitrobenzoic acid), 2,2-diphenyl-1-picrylhydrazyl (DPPH), 2,4,6-Tripyridyl-s-Triazine (TPTZ), Tris(hydroxymethyl)aminomethane, hydrochloric acid (36%), Trolox, Folin Ciocaltue’s reagent, gallic acid (95%), rutin (95%), quercetin (97%) and neostigmine bromide were purchased from Sigma Chemical Co. (St. Louis, MO, USA). Kojic acid (Alfa Aesar™, 99%), sodium phosphate monobasic monohydrate (ACROS Organics™, 98+%) and sodium hydrogen phosphate (Alfa Aesar™, 98+%) were provided by Fisher Scientific (Hampton, VA, USA). Ethanol absolute was purchased from Sodipro (Echirolles, France). All other chemicals and reagents used were analytical grade. The water used for the extraction and preparation of water-based solutions was Milli-Q grade (Millipore, Burlington, MA, USA).

### 3.2. Extraction Methods

#### 3.2.1. Production of Red Seaweed Extracts

To extract the broad spectrum of minor compounds from *Gelidium sesquipedale*, two methods were investigated: UAE and the traditional method to compare the extraction efficiency. UAE was optimized by studying different extraction conditions such as temperature, time, and solvent type (Table 5). Moreover, Table 5 shows the targeted bioactive or bioactivity evaluation of each algal extract.

##### Ultrasound-Assisted Extraction

UAE was carried out with an ultrasound bath (ULTR-3L2-001, Labbox Labware, S.L., Barcelona, Spain) with automatic control of time and temperature, an ultrasound frequency of 42 kHz and ultrasonic power of 100 W. Dried and milled red seaweed samples were dispersed in ethanol, or distilled water or aqueous ethanolic solutions (50% and 70% *v*/*v*) at a ratio of 1:20 (*w*/*v*). Different experiments were carried out using different parameters like temperature (room temperature (RT) and 40 °C), and time (15 and 30 min) (Table 5) [17]. After the treatment, samples were centrifuged at 4000× *g* for 5 min. The supernatants collected were stored in dark vessels at 4 °C until their analysis. All extraction procedures were performed in triplicate.

##### Conventional Solvent Extraction

Traditional method was carried out to compare the efficacy of conventional solvent extraction with the UAE conditions determined in this study. Seaweed samples were extracted following the method described by Ummat et al. [28], with minor modifications. Briefly, dried and milled red seaweed samples were dispersed in ethanol, or distilled water or aqueous ethanolic solutions (50% and 70% *v*/*v*) at a ratio of 1:20 (*w*/*v*), and the biomolecules of interest were extracted in an orbital shaker (RT, 200 rpm and for 4 h). After the treatment, samples were centrifuged at 4000× *g* for 5 min. The macroalgal pellet was re-extracted following the same procedure, and both supernatants were pooled together. Ethanol was evaporated in a rotary evaporator (Buchi R-100, BÜCHI Labortechnik AG, Switzerland) and the remaining aqueous fractions were freeze-dried. Extracts were re-dissolved in the same volume used for the ultrasound extracts. Samples were stored in dark vessels at 4 °C until their analysis. All extraction procedures were performed in triplicate.

#### 3.2.2. Production of Phycobiliproteins Extracts

The extraction of phycobiliproteins from *Gelidium sesquipedale* was done following different methods using phosphate buffer (0.1 M, pH 6.8) as a solvent: serial extraction (traditional), UAE, and a combination of UAE and maceration. Additionally, different parameters affecting the extraction of phycobiliproteins, such as ultrasonication and maceration times, were investigated (Table 6).

##### Serial Extraction

Serial extraction (traditional extraction method) was carried out to estimate the maximum extractable content of phycobiliproteins (R-phycoerythrin, R-PE and R-phycocyanin, R-PC) present in algal biomass. The traditional extraction was performed following a previously described protocol for *Gelidium pusillum* (Rhodophyta) [45] with minor modifications. Briefly, 2.5 g of *Gelidium sesquipedale* powder was macerated in 25 mL of phosphate buffer (0.1 M, pH 6.8) for 1 h at 4 °C under constant stirring. After the extraction time, samples were centrifuged at 4000× *g* for 12 min at 4 °C. The supernatant was collected for UV analysis and the pellet was submitted for another cycle of extraction with fresh solvent. This procedure was repeated until no detectable phycobiliproteins were extracted in the buffer (5 cycles) and the supernatants of each cycle were pooled together for spectrophotometric analysis. This process was used as a reference to estimate the maximum amount of phycobiliproteins that could be extracted. The serial extraction was done in triplicate.

##### Ultrasound-Assisted Extraction and Combination with Maceration

The UAE method was carried out as follows: 2.5 g of *Gelidium sesquipedale* powder were dispersed in 25 mL of phosphate buffer (0.1 M, pH 6.8) and extraction was done in an ultrasonic bath at room temperature using different times: 10 and 15 min. For the experiments using the combination of UAE and maceration, samples were treated as described before for the UAE method, followed by a maceration step. After ultrasonic treatment, samples were macerated in phosphate buffer (0.1 M, pH 6.8) for different times (45 min and 1 h) at 4 °C under constant stirring. After the treatments, samples were centrifugated at 4000 g for 12 min at 4 °C and the supernatants were collected for UV analysis. All extraction procedures were performed in triplicate.

### 3.3. Determination of Total Phenolic Content (TPC)

The TPC of seaweed extracts was determined according to the Folin–Ciocalteu method previously described [53,54], with slight modifications and using gallic acid (GA) as standard [55]. Briefly, in a 96-well plate (Thermo Scientific™ Multiskan Sky, Waltham, MA, USA), 20 µL of or serial standard solution was mixed with 100 µL of Folin–Ciocalteu reagent (10% in distilled water). After 5 min, 80 µL of 7.5% (*v*/*w*) sodium carbonate solution was added. The reaction mixture was incubated at room temperature and darkness for 30 min. The absorbance was measured at 750 nm. The calibration curve was prepared by using gallic acid ethanolic solutions and phenolic content was expressed as mg gallic acid equivalents per 100 g of dry algae (mg GAE/100 g).

### 3.4. Determination Total Flavonoid Content (TFC)

The TFC of seaweed extracts was determined by following the colorimetric method previously described using rutin as standard [56]. Briefly, in a 96-well plate (Thermo Scientific™ Multiskan Sky, USA), 20 µL of each extract were mixed with 20 µL of 10% aluminum chloride, 20 µL of 1 M potassium acetate and 180 µL of distilled water. The reaction mixture was incubated at room temperature and darkness for 30 min. The absorbance was measured at 415 nm. The calibration curve was prepared by using rutin methanolic solutions and flavonoid content was expressed as mg rutin equivalents per 100 g of dry algae (mg RE/100 g).

### 3.5. Identification and Quantification of Mycosporines and MAAs

#### 3.5.1. Chromatographic Conditions

Experiments were performed on a HILIC Osaka Soda Capcell Core PC column (2.1 × 150 mm, 2.7 μm, 90 Å) from BGB Analytics (Saint-Jean de Gonville, France). The mobile phases were: 5 mM ammonium acetate in water at pH 5.3 (A) and acetonitrile (B). The flow rate was fixed at 0.35 mL/min. The HPLC separation was carried out with the following gradient elution profile: 0–2 min, 10% B; 2–6 min, 20% B; 6–11 min, 20% B; 11–15.5 min, 80% B; 15.5–17.5 min, 80% B; 17.5–19.5 min, 10% B; 19.5–23 min, 10% B. A 15 µL aliquot of diluted extract was injected.

#### 3.5.2. Mycosporines and MAAs Profiling

The screening of mycosporines and MAAs in *Gelidium sesquipedale* extracts were based on the untargeted (HILIC)—Electrospray Orbitrap MS^2^/MS^3^ analysis developed by Parailloux et al. [15]. A set of eight characteristic fragment ions, and neutral and radical losses produced in the fragmentation pathways of mycosporines and MAAs were defined to flag unreported candidate-compounds.

For this purpose, LC-MS experiments were carried out with the following the ESI conditions: sheath gas 50 (arb), auxiliary gas 10 (arb), sweep gas 1 (arb), ion transfer tube and vaporizer temperature 350 °C, rf lens 50% and positive ionization voltage 3500 V. Full MS Orbitrap (OT) settings were: resolution 120,000, mass range *m*/*z* 150–500, dynamic exclusion 5 s and intensity threshold 2 × 10^4^. The ddMS2 OT settings were: resolution 60,000 for HCD70 MS/MS scans and 30,000 for HCD50 and CID30 MS/MS scans, isolation width 2 Da. The ddMS^3^ ion-trap (IT) settings were: scan rate 33,333 Da/s, peak width 0.5 FWHM, isolation width 2 Da. HCD70 and CID30 MS^2^ scans were used in parallel to generate both fragment ions, neutral losses and small radicals specific to the MAA compound class. Filtering criteria allowed the triggering of further ddMS^2^ HCD50 scans for structural elucidation of candidate-MAAs if characteristic fragment ions were detected in prior HCD70 MS² scans. Likewise, neutral and small radical losses were included in the second filter triggering a ddMS^3^ CID30 to confirm the detection of the candidate-MAAs found with the set of common fragment ions.

Data-treatment was performed on Compound Discoverer 3.2^TM^ software (ThermoFisher Scientific, Waltham, MA, USA) for the MAA annotation in every extract. An untargeted workflow was designed to sort out the compounds detected for which the signal intensity was above 1 × 10^5^ and the number of characteristic fragment ions was greater than or equal to four in their HCD70 MS² scan.

#### 3.5.3. Quantitative Analysis of MAAs

The recovery of MAAs for the triplicate of different extraction methods was evaluated by quantifying porphyra-334, shinorine, palythine and asterina-330. The amounts of the three first compounds were determined in algal extracts using the standard addition method. The spiked extracts were analyzed in targeted Selected Ion Monitoring (SIM) with the following parameters: isolation width 1 min; MS resolution 60,000. In absence of a standard, the quantification of asterina-330 was estimated assuming an electrospray MS response factor similar (within 10%) to the standards. All the spikes were carried in algal extracts diluted 100-fold.

### 3.6. Spectrophotometric Determination of Phycobiliproteins

Spectrophotometric determination of phycobiliproteins was performed following a previously described protocol for *Gelidium pusillum* [45]. R-PE and R-PC contents were estimated by measuring absorbance at 564 nm, 618 nm and 730 nm using a dual beam UV–visible spectrophotometer (Thermo Scientific™ Multiskan Sky, USA). The following equations were used for the estimation of phycobiliprotein contents:R − PE = 0.1247 [(A_564_ − A_730_) − 0.4583 (A_618_ − A_730_)](1)
R − PC = 0.154 (A_618_ − A_730_)](2)
Total amount = (R − PE) + (R − PC)(3)

All the experiments were carried out in triplicates.

### 3.7. Antioxidant Capacity Analysis

#### 3.7.1. DPPH Radical-Scavenging Assay

The antioxidant activity was measured through the determination of the radical scavenging activity using 2,2- diphenyl-1-picrylhydrazyl (DPPH) [57]. Briefly, 250 μL of 8.66 × 10^−5^ M DPPH methanolic solution was added to 50 μL of sample/standard in a 96-well plate (Thermo Scientific™ Multiskan Sky, USA) and incubated in the dark at room temperature for 30 min. The absorbance was measured at λ 517 nm. Distilled water was used as a blank. Each standard or sample solution was run in triplicate. The ability to scavenge the DPPH radical was calculated using the follow equation:Scavenging effect (%) = [1 − (A_sample_ − A_sample blank_)/A_control_] × 100(4)
where the A_control_ is the absorbance of the control (DPPH solution without sample), the A_sample_ is the absorbance of the test sample (DPPH solution plus test sample), and the A_sample blank_ is the absorbance of the sample only (sample without DPPH solution). Trolox standard was used to generate a standard curve and results were expressed as mg trolox equivalents (TE) per 100 g of dry algae (mg TE/100 g) [28].

#### 3.7.2. Ferric Reducing Antioxidant Power (FRAP) Assay

The antioxidant activity of seaweeds extracts was measured using FRAP assay according to the Benzie and Strain [58] method, with some modifications [53]. First, the working FRAP solution was prepared mixing 300 mM acetate buffer (pH 3.6), 10 mM TPTZ 40 mM HCl and 20 mM FeCl_3_ in a ratio 10:1:1 (*v*/*v*/*v*). The FRAP solution was freshly prepared and warmed at 37 °C for 10 min. Then, 20 μL of test sample solution was dispensed to each microplate well and the reaction was initiated by the addition of 200 μL of the FRAP working solution. The reaction mixture was incubated at room temperature and in darkness for 10 min. Finally, the absorbance was measured at 593 nm. Trolox standard was used to make a standard curve and results were expressed as mg trolox equivalents (TE) per 100 g of dry algae (mg TE/100 g).

### 3.8. Anti-Enzymatic Activities

#### 3.8.1. Acetylcholinesterase (AChE) Inhibition Assay

Acetylcholinesterase (AChE, Type-VI-S, EC 3.1.1.7, 222 U/mg) inhibitory activity of extracts were determined according to previously described Ellman’s colorimetric method [59]. Acetylthiocholine iodide was employed as the substrate to assay the inhibition of AChE. The reaction mixture contained 130 μL of 100 mM sodium phosphate buffer (pH 8.0), 20 μL of test sample solution and 20 μL of AChE (0.36 U/mL), which were mixed and incubated for 15 min at 25 °C. The reaction was then initiated via the addition of 40 μL of the following mixture (freshly prepared): 20 μL 0.5 mM DTNB (5,5′-dithiobis (2-nitrobenzoic acid) and 20 μL acetylthiocholine iodide (0.71 mM). The hydrolysis of acetylthiocholine iodide was monitored by following the formation of yellow 5-thio-2-nitrobenzoate anion at 412 nm every 10 sec for 10 min using a 96-well microplate reader under a constant temperature of 25 °C, which resulted from the reaction of 5–50-thiobis-2-nitrobenzoic acid with thiocholine, released by the enzymatic hydrolysis of acetylthiocholine iodide. The percent inhibition of acetylcholinesterase enzyme was calculated using the equation:% Inhibition = [(ΔAbs/min_control_ − ΔAbs/min_sample_)/ΔAbs/min_control_] × 100(5)
where, Abs_control_ is the absorbance of the assay using the buffer instead of inhibitor (sample) and Abs_sample_ is the absorbance of the sample extracts. Neostigmine bromide was used as the positive control [49]. Phosphate buffer was used as the blank. Each standard or sample solution was analyzed in triplicate. The concentration of the extracts which caused 50% inhibition of the tyrosinase activity (IC50) was calculated by nonlinear regression analysis.

#### 3.8.2. Elastase Inhibition Assay

The elastase inhibition of seaweeds extracts was investigated in TRIS buffer solution following the method of Eun et al. [60], with some modifications [61]. Briefly, 100 µL of 0.1 M Tris-HCl buffer solution (pH 8.0), 25 µL of elastase (1 U/mL in TRIS buffer) and 20 µL sample extracts were incubated for 15 min at 25 °C, before adding the substrate to begin the reaction. After incubation time, 40 µL of 0.5 mM N-Succinyl-Ala-Ala-Ala-p-nitroanilide (AAAPVN) solution in water was added. Following this, absorbance at 410 nm was monitored for 20 min using a microplate reader under a constant temperature of 25 °C. Finally, elastase inhibition was calculated in percentage using the equation:% Inhibition = [(ΔAbs/min_control_ − ΔAbs/min_sample_)/ΔAbs/min_control_] × 100(6)
where, Abs_contro_l is the absorbance of the assay using the buffer instead of inhibitor (sample) and Abs_sample_ is the absorbance of the sample extracts. Quercetin was used as the positive control [61]. Tris-HCl buffer was used as the blank. Each standard or sample solution was analyzed in triplicate. The concentration of the extracts which caused 50% inhibition of the tyrosinase activity (IC50) was calculated by nonlinear regression analysis.

#### 3.8.3. Tyrosinase Inhibition Assay

Tyrosinase inhibitory assay was performed according to the method previously described by [61], with some modifications using 3,4-Dihydroxy-L-phenylalanine (L-DOPA) as substrate. A volume of 20 µL of sample, 20 µL of mushroom tyrosinase solution (100 U/mL in phosphate buffer) and 80 µL of phosphate buffer (pH = 6.8) were mixed and pre-incubated at 37 °C for 5 min. Then, 90 µL of L-DOPA (2 mg/mL water) was added. The formation of dopachrome was immediately monitored for 20 min at 475 nm in a microplate reader under constant temperature of 37 °C. The percent inhibition of tyrosinase enzyme was calculated using the equation:% Inhibition = [(ΔAbs/min_control_ − ΔAbs/min_sample_)/ΔAbs/min_control_] × 100(7)
where, Abs_control_ is the absorbance of the assay using the buffer instead of the inhibitor (sample) and Abs_sample_ is the absorbance of the sample extracts. Kojic acid was used as the positive control. Phosphate buffer was used as the blank. Each standard or sample solution was analyzed in triplicate. The concentration of the extracts which caused 50% inhibition of the tyrosinase activity (IC50) was calculated by nonlinear regression analysis.

### 3.9. Statistical Analysis

All the experiments were carried out in triplicate. Results are expressed as mean ± standard deviation. Statistical analysis was performed using JASP version 0.14.1 (University of Amsterdam, Netherlands). The effect of extraction method, extraction time and solvents on the bioactives recovery and associated antioxidant and anti-enzymatic activities were analyzed using ANOVA followed by Tukey’s HSD post-hoc tests. In all cases, differences were considered statistically significant at *p* < 0.05.

## 4. Conclusions

The present study provides relevant results for the development of eco-friendly approaches to reach a superior valorization of the red algae *Gelidium sesquispedale* by extracting high-valuable bioactives such as phenolic and flavonoid compounds, mycosporine-like amino acids, and phycobiliproteins. The green methods developed using ultrasound-assisted extraction and the combination of green solvents showed a promising alternative to efficiently extract a broad spectrum of minor compounds, achieving comparable extraction yields to traditional methods in very short times (15 to 30 min vs. 8 h for the conventional method). Additionally, produced algal extracts exhibited interesting antioxidant and anti-enzymatic activities for potential applications in medical and/or cosmetic products. Thus, this study shows the importance of eco-friendly methods to studying novel properties of algal biomass to provide new valorization routes. The proposed extraction methods not only open new possibilities for the commercial use of *Gelidium sesquipedale,* but also for the valorization of different algae species since the techniques established can be easily adapted, proving the usefulness of green strategies to extract bioactive compounds from algal biomass.

## Figures and Tables

**Figure 1 marinedrugs-19-00574-f001:**
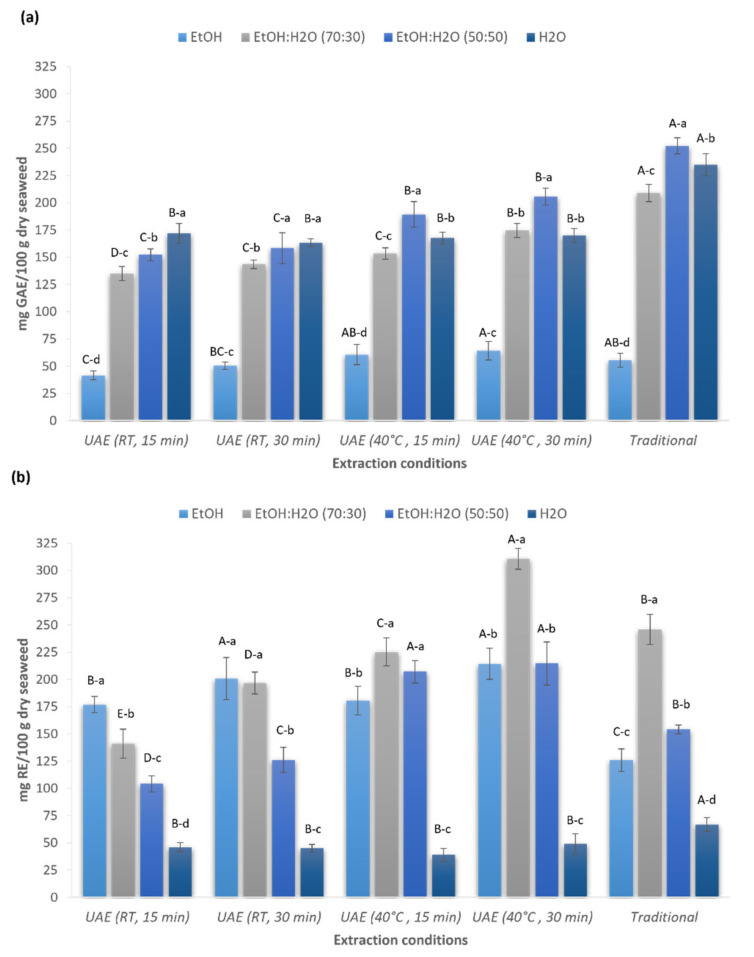
Effect of ultrasound-assisted extraction process parameters (time, temperature, and solvents) on total polyphenol (**a**) and total flavonoid (**b**) content of *Gelidium sesquipedale* seaweed. TPC (total phenolic content) and TFC (total flavonoid content) are expressed as mg gallic acid equivalents (GAE)/100 g dry algae, and mg quercetin equivalents (QE)/100 g dry algae, respectively. Data are shown as mean ± SD (n = 9). Capital letters indicate statistically significant differences in extraction conditions and lowercase letters indicate statistically significant differences in solvents (one-way ANOVA with post-hoc Tukey, *p* < 0.05).

**Figure 2 marinedrugs-19-00574-f002:**
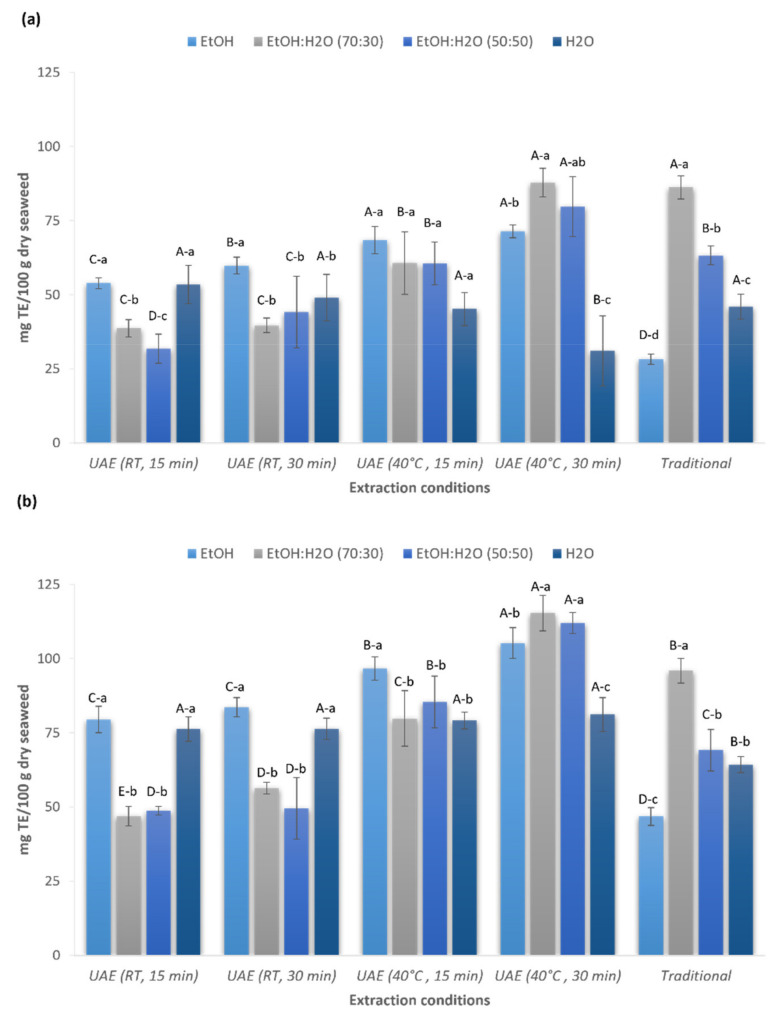
Antioxidant capacity measured as 1,1-diphenyl-2-picryl-hydrazyl (DPPH) activity (**a**) and ferric reducing antioxidant power (FRAP) (**b**) of *Gelidium sesquipedale* extracts using UAE and conventional solvent extraction techniques. DPPH and FRAP: expressed as mg trolox equivalent (TE)/g of 100 g dry algae. Data are shown as mean ± SD (n = 9). Capital letters indicate statistically significant differences in extraction conditions, and lowercase letters indicate statistically significant differences in solvents (one-way ANOVA with post-hoc Tukey, *p* < 0.05).

**Table 1 marinedrugs-19-00574-t001:** Profiling of candidate-MAAs in *Gelidium sesquipedale* water extracts obtained using the traditional extraction method and ultrasound-assisted extraction (UAE) applying different extraction times (15 and 30 min) and temperatures (RT and 40 °C).

										Area Max (10^6^)				
										Traditional	Ultrasound-Assisted Extraction (UAE)			
Name	ISF ^1^	Formula	Mass Error (ppm)	M_mi_ ^2^ (Da)	[M + H]^+^ (*m*/*z*)	RT [min]	CFI ^3^ (/8)	FISh Score ^4^ (%)	Fragment Ions ^5^ (MS²)		RT 15 min	40 °C 15 min	RT 30 min	40 °C 30 min
Asterina-330		C_12_ H_20_ O_6_ N_2_	−N.11	288.1318	289.1390	10.83	8	40	274.1157; 230.1260; 212.1155; 186.0998	391 ± 78	318 ± 62	230 ± 14	218 ± 21	62.3 ± 46
Asterina-330	[(M + H)—(CH_3_)]	C_11_ H_17_ O_6_ N_2_	−N.7	273.1087	274.1159	10.83	8	-	230.1260; 212.1155; 186.0998	7.4 ± 0.7	4.9 ± 0.5	5.23 ± 0.5	4.61 ± 0.5	2.42 ± 0.2
Aplysiapalythine A		C_13_ H_22_ O_6_ N_2_	0.35	302.1479	303.1550	9.76	8	50	288.1316; 244.1416; 186.0998	1.22 ± 0.3	2.33 ± 0.03	2.71 ± 0.8	1.94 ± 0.03	0.9 ± 0.08
Porphyra-334		C_14_ H_22_ O_8_ N_2_	−0.15	346.1376	347.1446	8.73	8	37	303.1187; 288.1316; 244.1417; 227.1026; 209.0920	7.25 ± 0.9	12.4 ± 3.3	7 ± 1	8.12 ± 1.4	0.65 ±0.4
Palythine		C_10_ H_16_ O_5_ N_2_	−N.78	244.1057	245.1131	10.14	8	45	230.0897; 209.0921; 86.0998	176 ± 35	131 ± 58	75 ± 4.5	95.4 ± 5.5	0.16 ± 0.02
Aplysiapalythine C	[(M + H)—(CO)]	C_10_ H_18_ O_4_ N_2_	0.21	230.1267	231.1340	10.84/15.05	7	-	216.1104; 172.0840	4.26 ± 2.2	2.93 ± 0.9	2.57 ± 0.2	2.73 ± 0.4	2.18 ± 0.3
Unknown		C_12_ H_20_ O_7_ N_2_	−N.36	304.1269*	305.1342	9.13	7	-	287.1238; 275.1238; 245.1132; 230.0898; 86.0998	2.18 ± 0.2	1.91 ± 0.2	1.03 ± 0.06	2.05 ± 0.3	< 0.01
Aplysiapalythine B		C_12_ H_20_ O_5_ N_2_	−N.43	272.1371	273.1443	8.58	7	55	258.1208; 214.1310; 183.1128; 165.1021	2.83 ± 0.6	1.78 ± 0.01	1.73 ± 0.4	1.80 ± 0.1	< 0.01
Shinorine		C_13_ H_20_ O_8_ N_2_	−N.05	332.1216	333.1289	9.13	5	34	318.1058; 303.1187; 255.0973; 274.1159; 230.1260; 186.0998	25 ± 2.4	50 ±16	27 ± 0.3	33 ± 3	0.2 ± 0.01
Aplysiapalythine C		C_11_ H_18_ O_5_ N_2_	0.23	258.1216	259.1289	10.82	4	48	241.1182; 231.1337; 191.0815	1.78 ± 0.5	1.79 ± 0.13	1.79 ± 0.7	2.37 ± 0.25	1.63 ± 0.2

^1^ In-Source-Fragmentation (ISF). ^2^ Monoisotopic mass (Mmi). ^3^ The number of characteristic fragment ions (CFI) was determined in HCD70 MS² scans for all the detected masses. Compounds displaying at least four characteristic fragment ions are considered as potential candidate-MAAs. ^4^ For every selected exact mass, a Fish score (%) was calculated to confirm the structural annotation given with the untargeted method. The score indicates the number of total experimental fragment ions matching with those found in silico. ^5^ Structural elucidation was carried on the basis of the most intense fragment ions detected in CID30 and HCD50 MS² scans and produced in the course of the fragmentation pathways of MAAs inducing the appearance of neutral and radical losses (CH_3_^−^, H_2_O, CO_2_, CH_3_O, C_2_H_4_O, C_3_H_6_O), for the exact mass **304.1269.** * Da, additional MS² fragmentation analysis was performed to complete fragmentation data.

**Table 2 marinedrugs-19-00574-t002:** Quantification of asterina-330, porphyra-334, palythine and shinorine in *Gelidium sesquipedale* water extracts obtained using the traditional method and ultrasound-assisted extraction (UAE), applying different extraction times (15 and 30 min) and temperatures (RT and 40 °C).

	MAAs Content (mg/100 g dw)				
Extraction Method	*Asterina-330*	*Porphyra-334*	*Palythine*	*Shinorine*	*Σ MAAs*
Traditional method	832.4 ± 166.3 ^a^	37.4 ± 4.6 ^b^	374.3 ± 74.6 ^a^	127.9 ± 12.3 ^b^	1372 ± 253.8 ^a^
UAE RT 15 min	676.2 ± 131.7 ^ab^	63.6 ± 16.8 ^a^	279.5 ± 123.9 ^ab^	252.5 ± 82.1 ^a^	1271 ± 352.9 ^ab^
UAE RT 30 min	468.4 ± 44.8 ^bc^	42.2 ± 7.0 ^ab^	205.2 ± 11.8 ^b^	171.0 ± 15.5 ^a^	886 ± 74.2 ^b^
UAE 40 °C 15 min	494.4 ± 30.7 ^bc^	36.5 ± 5.5 ^b^	161.6 ± 9.6 ^c^	139.5 ± 1.5 ^b^	832 ± 43.9 ^b^
UAE 40 °C 30 min	134.7 ± 98.0 ^d^	3.8 ± 2.0 ^c^	0.94 ± 0.04 ^d^	1.31 ± 0.07 ^c^	144 ± 97.9 ^c^

Data are shown as mean ± SD (n = 3); content of mycosporine-like amino acids (MAAs) is expressed as mg of MAAs/100 g dry algae; lowercase letters indicate statistically significant differences in columns (one-way ANOVA with post-hoc Tukey, *p* < 0.05).

**Table 3 marinedrugs-19-00574-t003:** Extraction of phycobiliproteins from *Gelidium sesquipedale* using different approaches and evaluation of the antioxidant capacity of produced extracts.

Extraction Method	R-PE Content (mg/100 g)	R-PC Content (mg/100 g)	Total Content (mg/100 g)	Extraction Efficiency (%)	DPPH (mg TE/100 g)	FRAP (mg TE/100 g)
*Traditional*						
Serial extraction (5 h)	97.1 ± 2.4 ^a^	50.2 ± 1.6 ^a^	147.3 ± 3.2 ^a^	100	34.9 ± 5.5 ^d^	13.8 ± 1.4 ^d^
*Ultrasound (UAE)*						
UAE 10 min	37.5 ± 1.0 ^d^	17.4 ± 0.7 ^c^	54.7 ± 1.6 ^d^	37	41.4 ± 3.3 ^bc^	21.7 ± 0.2 ^b^
UAE 15 min	37.6 ± 1.3 ^d^	16.5 ± 0.5 ^c^	54.1 ± 2.1 ^d^	37	38.5 ± 2.5 ^cd^	17.3 ± 1.6 ^c^
*Ultrasound + maceration (Mac)*						
UAE 15 min + Mac 45 min	48.3 ± 1.5 ^c^	24.0 ± 1.4 ^b^	72.4 ± 0.5 ^c^	49	48.9 ± 2.5 ^a^	23.5 ± 1.1 ^a^
UAE 15 min + Mac 1 h	52.4 ± 1.3 ^b^	25.7 ± 0.5 ^b^	77.9 ± 1.6 ^b^	53	45.6 ± 2.8 ^ab^	22.6 ± 0.8 ^ab^

Data are shown as mean ± SD (n = 3); phycobiliproteins (R-phycoerythrin, R-PE and R-phycocyanin, R-PC) contents are expressed as mg of phycobiliproteins/100 g dry algae; DPPH and FRAP values are expressed as mg trolox equivalent (TE)/100 g dry algae (n = 9). Lowercase letters indicate statistically significant differences in columns (one-way ANOVA with post-hoc Tukey, *p* < 0.05).

**Table 4 marinedrugs-19-00574-t004:** Acetylcholinesterase (AChE), tyrosinase and elastase inhibitory activities of aqueous red seaweed and phycobiliproteins extracts using UAE and conventional solvent extraction techniques.

Samples	AChE Assay IC_50_ (mg/mL)	Tyrosinase Assay IC_50_ (mg/mL)	Elastase Assay IC_50_ (mg/mL)
*Red seaweed extracts (water)*			
Traditional method	36.3 ± 3.1 ^a^	NI	NI
UAE RT 15 min	56.6 ± 1.5 ^b^	NI	NI
UAE RT 30 min	59.0 ± 3.1 ^b^	NI	NI
UAE 40 °C 15 min	68.2 ± 0.6 ^c^	NI	NI
UAE 40 °C 30 min	76.1 ± 6.2 ^c^	NI	NI
*Phycobiliproteins extracts*			
Serial extraction (5 h)	> 100 ^e^	> 100 ^b^	> 100 ^b^
US 15 min + Mac 45 min	94.3 ± 0.2 ^d^	86.5 ± 0.5 ^a^	87.4 ± 0.4 ^a^
*Positive standards*			
Neostigmine bromide	0.06 ± 0.01	NT	NT
Kojic acid	NT	0.05 ± 0.01	NT
Quercetin	NT	NT	0.22 ± 0.01

Data are shown as mean ± SD; IC_50_ values represent the mean standard error of three parallel measurements; lowercase letters indicate statistically significant differences in columns (one-way ANOVA with post-hoc Tukey, *p* < 0.05); US = ultrasound; Mac = maceration; NT = not tested; NI = no inhibition.

**Table 5 marinedrugs-19-00574-t005:** Experimental design for the extraction of different bioactive compounds and bioactivity evaluation of *Gelidium sesquipedale* using ultrasound-assisted extraction (UAE) and traditional extraction methods.

	Extraction Conditions						Targeted Bioactive Compound and/or Bioactivity Evaluation
Extraction Approach	Temperature	Time	Solvents				
			Ethanol	Ethanol:Water (70:30 v/v)	Ethanol:Water (50:50 v/v)	Water	
Traditional method (n = 3)	RT	8 h	x	x	x		phenolic compounds, flavonoids, and antioxidant properties
	RT	8 h				x	identification and quantification of MAAs and anti-enzymatic activities
UAE RT 15 min (n = 3)	RT	15 min	x	x	x		phenolic compounds, flavonoids, and antioxidant properties
	RT	15 min				x	identification and quantification of MAAs and anti-enzymatic activities
UAE RT 30 min (n = 3)	RT	30 min	x	x	x		phenolic compounds, flavonoids, and antioxidant properties
	RT	30 min				x	identification and quantification of MAAs and anti-enzymatic activities
UAE 40 °C 15 min (n = 3)	40 °C	15 min	x	x	x		phenolic compounds, flavonoids, and antioxidant properties
	40 °C	15 min				x	identification and quantification of MAAs and anti-enzymatic activities
UAE 40 °C 30 min (n = 3)	40 °C	30 min	x	x	x		phenolic compounds, flavonoids, and antioxidant properties
	40 °C	30 min				x	identification and quantification of MAAs and anti-enzymatic activities

The extraction was carried out in triplicate for each group and every extract replicate was analyzed in triplicate.

**Table 6 marinedrugs-19-00574-t006:** Experimental design for the extraction of phycobiliproteins and bioactivity evaluation from *Gelidium sesquipedale* using ultrasound-assisted extraction (UAE) and traditional extraction methods.

	Extraction Conditions				Targeted Bioactive Compound and/or Bioactivity Evaluation
Extraction Approach	Maceration		Ultrasound		
	Temperature	Time	Temperature	Time	
*Traditional*					
Serial extraction (n = 3)	-	-	−4 °C	5 h	phycobiliproteins, antioxidant properties and anti-enzymatic activities
*Ultrasound (UAE)*					
UAE 10 min (n = 3)	RT	10 min	-	-	phycobiliproteins and antioxidant properties
UAE 15 min (n = 3)	RT	15 min	-	-	phycobiliproteins and antioxidant properties
*Ultrasound + maceration (UAE + Mac)*					
UAE 15 min + Mac 45 min (n = 3)	RT	15 min	−4 °C	45 min	phycobiliproteins, antioxidant properties and anti-enzymatic activities
UAE 15 min + Mac 1 h (n = 3)	RT	15 min	−4 °C	1 h	phycobiliproteins and antioxidant properties

The extraction was carried out in triplicate for each group and every extract replicate was analyzed in triplicate.

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
