# Peer review of "Valorization of the Red Algae Gelidium sesquipedale by Extracting a Broad Spectrum of Minor Compounds Using Green Approaches"

_marinedrugs, 2021, doi:10.3390/md19100574_

Round 1

Reviewer 1 Report

This manuscript showed a new process for the extraction of minor compounds of the red algae Gelidium sesquipedale by using eco-friendly extraction approaches (ultrasound-assisted extraction, UAE) in combination with green solvents. UAE process shows good results for extraction of total phenolics, flavonoids, mycosporine-like amino acids (MAAs), phycobiliproteins (R-Phycoerythrin (R-PE) and R-Phycocyanin (R-PC) and anti-enzymatic activities (acetylcholinesterase (AChE), tyrosinase, and  elastase). Therefore, it is a promissory process.

Therefore, the content of this manuscript is pertinent to this journal, however, the following points need to be addressed before acceptance in this journal:

  • Could the authors indicate the level of cost and energy consume of UAE process in comparison with traditional process?
  • How UAE process affect agar extraction from Gelidium sesquipedale?
  • How conditions of time, temperature, and solvents for UAE process were selected?
  • Are there other type of phycobiliproteins in Gelidium sesquipedale, for instance Allophycocyanin (APC) and Phycoerythrocyanin (PEC)?

Figures 1 and 2. B Please Include nomenclature of each bar.

Author Response

The new revised version of the manuscript ID-marinedrugs-1407826 has been improved following comments and suggestions from reviewers. The authors would like to thank reviewers 1,2 and 3 for taking the time an effort necessary to review the earlier version of the manuscript.

Response to Reviewer 1 Comments

This manuscript showed a new process for the extraction of minor compounds of the red algae Gelidium sesquipedale by using eco-friendly extraction approaches (ultrasound-assisted extraction, UAE) in combination with green solvents. UAE process shows good results for extraction of total phenolics, flavonoids, mycosporine-like amino acids (MAAs), phycobiliproteins (R-Phycoerythrin (R-PE) and R-Phycocyanin (R-PC) and anti-enzymatic activities (acetylcholinesterase (AChE), tyrosinase, and  elastase). Therefore, it is a promissory process.

Therefore, the content of this manuscript is pertinent to this journal, however, the following points need to be addressed before acceptance in this journal:

  • Could the authors indicate the level of cost and energy consume of UAE process in comparison with traditional process

Reviewer 1 raises an important point here. It would be interesting to evaluate the real cost and energy use of both methods. However, for the moment, the authors can only claim the advantages of UAE process as a faster alternative in comparison with the traditional method. There are several studies regarding the environmental profiles and life cycle assessment of extraction processes (for example Barjoveanu et al., 2020 or Vauchel et al., 2018). The evaluation of energy cost is one of the main actual concerns of our research group, so we hope that we can include them together with the life cycle assessment in our next papers.

References:

Barjoveanu, G. et al. Life cycle assessment of polyphenols extraction processes from waste biomass. Scientific Reports, 10, 13632 (2020). https://doi.org/10.1038/s41598-020-70587-w

Vauchel, P. et al. LCA of ultrasound-assisted extraction of polyphenols from chicory grounds under different operational conditions. Journal of Cleaner Production, 196, 1116-1123, (2018) https://doi.org/10.1016/j.jclepro.2018.06.042

  • How UAE process affect agar extraction from Gelidium sesquipedale?

The application of ultrasound for the extraction of agar-agar from Gelidium sesquipedale has been already described for other research groups (Martínez-Sanz et al., 2019), showing that UAE not only allowed to obtain an agar-agar with good quality that also ultrasounds allowed to reduce 4-fold the extraction time in comparison with the traditional hot water extraction. However, a more interesting approach, in which the research group is now working, is to evaluate the complete valorization of Gelidium biomass using the UAE methods established herein.

References:

Martínez-Sanz, M. et al. Production of unpurified agar-based extracts from red seaweed Gelidium sesquipedale by means of simplified extraction protocols, Algal Research, 38, 101420 (2019) https://doi.org/10.1016/j.algal.2021.102254

  • How conditions of time, temperature, and solvents for UAE process were selected?

The extraction conditions for the UAE process were based on preliminary studies of the research group and previously published studies of one of the members of the group (Castejón et al., 2018), but for extracting different bioactive compounds, like omega-3 fatty acids. In the revised version of the manuscript, the reference has been added (section 3.2.1.1. Ultrasound assisted extraction).

References:

Castejón, N et al. Alternative Oil Extraction Methods from Echium Plantagineum L. Seeds Using Advanced Techniques and Green Solvents. Food Chemistry, 244, 75–82. (2018) https://doi.org/10.1016/j.foodchem.2017.10.014

  • Are there other type of phycobiliproteins in Gelidium sesquipedale, for instance Allophycocyanin (APC) and Phycoerythrocyanin (PEC)?

In this study, the authors only focused on the determination of the main phycobiliproteins from red algae, namely R-Phycoerythrin (R-PE) and R-Phycocyanin (R-PC). Unfortunately, the method used only allowed to estimate the content of R-PE and R-PC. Nevertheless, it would be very interesting to analyze the algal extracts in more detail to investigate if other phycobiliproteins are present.

  • Figures 1 and 2. B Please Include nomenclature of each bar.

Done. In the revised version of the manuscript, Figures 1 (b) and 2 (b) have the appropriate legend.

Reviewer 2 Report

For valorization of seaweeds resources, it is significant to conduct the "green" extraction for full use of biomass, for ecological and environmental consideration, the applied UAE methodologies will potentially choice in the large extraction in future. 

It is needs to make minor revisions to the MS in a more concise style, especially to surmmarize the extration rate compared with the traditional methods to the these traget compounds.  

Author Response

The new revised version of the manuscript ID-marinedrugs-1407826 has been improved following comments and suggestions from reviewers. The authors would like to thank reviewers 1,2 and 3 for taking the time an effort necessary to review the earlier version of the manuscript.

Response to Reviewer 2 Comments

For valorization of seaweeds resources, it is significant to conduct the "green" extraction for full use of biomass, for ecological and environmental consideration, the applied UAE methodologies will potentially choice in the large extraction in future. 

It is needs to make minor revisions to the MS in a more concise style, especially to surmmarize the extration rate compared with the traditional methods to the these traget compounds.  

Reviewer 2 is right; the discussion of the MS could be simplified to give the reader a better understanding. In the revised version of the manuscript, the discussion of the MS data had been modified to clarify this part of the article.

Reviewer 3 Report

Dear authors,

The manuscript presents good results and shows state-of-the art methods for example extraction using sonochemistry concept and LCMS/MS to detect the presence of major and minor compounds. The experiment is also equipped by statistical analyses. I recommend to accept this manuscript after some minor corrections. I have one questions: Why don't authors take LC-MS/MS in negative mode? This anticipates the presence of sulfate or carboxcylic or other related functional groups in the algae.   

Author Response

The new revised version of the manuscript ID-marinedrugs-1407826 has been improved following comments and suggestions from reviewers. The authors would like to thank reviewers 1,2 and 3 for taking the time an effort necessary to review the earlier version of the manuscript.

Response to Reviewer 3 Comments

Dear authors,

The manuscript presents good results and shows state-of-the art methods for example extraction using sonochemistry concept and LCMS/MS to detect the presence of major and minor compounds. The experiment is also equipped by statistical analyses. I recommend to accept this manuscript after some minor corrections. I have one questions: Why don't authors take LC-MS/MS in negative mode? This anticipates the presence of sulfate or carboxcylic or other related functional groups in the algae.   

Although negative mode has been preferentially used for the study of dicarboxylic MAAs, decarboxylation phenomenon has been also observed in the fragmentation routes of MAAs in positive mode (Cardozo et al. 2008, 2009). More recently, the presence of sulfate in the MAA structure has been proved by the release of SO3 (Δ mass 79.957 Da) in their fragmentation pathways in positive ionization mode as well (Parailloux et al. 2020). Because of their zwitterionic properties, their abundance in amino- and imino-groups and their structural diversity, MAAs are more easily ionizable in positive mode. That is the reason why a positive ionization has been selected as the best compromise for the full coverage of the MAA contents in the present study, whatever the nature of their substituents, as part of the present MS/MS analysis (Whitehead and Edges, 2002, 2003; Cardozo et al. 2008).

References:

- Cardozo et al. (2008), A theoretical and mass spectrometry study of the fragmentation of mycosporine-like amino acids, International Journal of Mass Spectrometry. https://doi.org/10.1016/j.ijms.2008.02.014

- Cardozo et al. (2009), A Fragmentation Study of Di-Acidic Mycosporine-like Amino Acids in Electrospray and Nanospray Mass Spectrometry, Journal of the Brazilian Chemical Society. https://doi.org/10.1590/S0103-50532009000900009

- Whitehead and Edges (2002), Analysis of mycosporine-like amino acids in plankton by liquid chromatography electrospray ionization mass spectrometry, Marine Chemistry. https://doi.org/10.1016/S0304-4203(02)00096-8

- Whitehead and Edges (2003), Electrospray ionization tandem mass spectrometric and electron impact mass spectrometric characterization of mycosporine-like amino acids. https://doi.org/10.1002/rcm.1162

- Parailloux et al. (2020), Untargeted Analysis for Mycosporines and Mycosporine-Like Amino Acids by Hydrophilic Interaction Liquid Chromatography (HILIC)—Electrospray Orbitrap MS2/MS3, Antioxidants. https://doi.org/10.3390/antiox9121185